# Impact of delivery mode-associated gut microbiota dynamics on health in the first year of life

Marta Reyman [1,2], Marlies A. van Houten[2], Debbie van Baarle[3], Astrid A.T.M. Bosch[1], Wing Ho Man[1,2], Mei Ling J.N. Chu[1], Kayleigh Arp[1], Rebecca L. Watson[4], Elisabeth A.M. Sanders[1,3], Susana Fuentes [3,5] & Debby Bogaert[1,4,5]*

The early-life microbiome appears to be affected by mode of delivery, but this effect may depend on intrapartum antibiotic exposure. Here, we assess the effect of delivery mode on gut microbiota, independent of intrapartum antibiotics, by postponing routine antibiotic administration to mothers until after cord clamping in 74 vaginally delivered and 46 caesarean section born infants. The microbiota differs between caesarean section born and vaginally delivered infants over the first year of life, showing enrichment of *Bifidobacterium* spp., and reduction of *Enterococcus* and *Klebsiella* spp. in vaginally delivered infants. The microbiota composition at one week of life is associated with the number of respiratory infections over the first year. The taxa driving this association are more abundant in caesarean section born children, providing a possible link between mode of delivery and susceptibility to infectious outcomes.

[1] Department of Paediatric Immunology and Infectious Diseases, Wilhelmina Children's Hospital of University Medical Centre, Utrecht, the Netherlands. [2] Spaarne Gasthuis Academy Hoofddorp and Haarlem, Hoofddorp, The Netherlands. [3] National Institute for Public Health and the Environment, Bilthoven, The Netherlands. [4] Medical Research Council/University of Edinburgh Centre for Inflammation Research, Queen's Medical Research Institute, University of Edinburgh, Edinburgh, UK. [5] These authors contributed equally: Susana Fuentes, Debby Bogaert. *email: D.Bogaert@ed.ac.uk

The impact of the human microbiome on health is becoming increasingly clear, with perturbations being associated with various (immune) disorders ranging from allergies and obesity to inflammatory bowel disease[1]. The gastrointestinal (GI) tract is of particular relevance to human health, as it contains the majority and most diverse set of human commensal bacteria[2]. Early-life gut microbiota development is crucial for a balanced priming of the immune system, which occurs early in life in the so-called window of opportunity[3]. Mode of delivery is considered a critical influential factor on gut microbiota development. Birth by caesarean section (CS) has been associated with adverse effects on immune development, predisposing to infections, allergies, and inflammatory disorders[4–6]. The rising incidence of CS births is alarming, reaching up to 40.5% of all births in some countries[7]. Research into the impact of CS birth on microbiota and health argues that results may be largely affected by intrapartum antibiotics[8]. Furthermore, the diminished success of breastfeeding after CS[9] adds to alterations in normal microbiota development[10].

In our study, we investigate a cohort of 120 healthy children not directly exposed to intrapartum antibiotics (Microbiome Utrecht Infant Study [MUIS]). In case of CS delivery, mothers were administered perioperative prophylaxis only after clamping of the umbilical cord, making it possible to focus on the independent effects of delivery mode on the gut microbiota. The infants were intensively sampled directly after birth throughout the first year of life. Fecal samples were collected from mothers, and a broad scale of clinical data and environmental and lifestyle characteristics was obtained for all participants. Primarily, we characterized the infant fecal microbiota composition and dynamics over the first year of life and assess, next to delivery mode, the effect of multiple variables, such as feeding type and early-life antibiotic use. Secondarily, we assess the effect of the observed delivery mode-induced gut microbiota alterations on infant health.

Here, we report on differences in the fecal microbiota between CS and vaginally delivered (VD) infants over the first year of life, independent of maternal antibiotics. In VD infants, we find evidence for fecal microbiota seeding from mother to infant and a more stable microbiota development in early-life compared with CS infants. Regarding specific taxa, VD infants show, among others, an enrichment of health-associated *Bifidobacterium* spp., and a reduction of potentially pathogenic *Enterococcus* and *Klebsiella* spp. The overall microbiota composition differs most pronouncedly between the delivery mode groups at 1 week of age. At this early timepoint, the microbiota composition is associated with the number of respiratory infections (RIs) a child will suffer from in the first year of life. Taxa strongly associated with more RIs are more abundant in CS children, providing a possible link between mode of delivery and susceptibility to infectious outcomes.

## Results
**Population characteristics**. In our study population, 74 children were VD, and 46 children were born by CS. Of those, 36 (78%) were born by planned, and 10 (22%) by emergency CS. There were two cases of pre-/intrapartum antibiotics, one in each delivery mode group, indicated for maternal fever, both of which were included in the analyses. All but three children were born in the hospital. Baseline characteristics and cumulative disease parameters over the first year of life of all children, stratified by mode of delivery, are shown in Table 1. Clinical variables were evenly distributed over both groups with the exception of gestational age (two-sample $t$ test, $p = 0.003$), duration of ruptured membranes ($p = 0.019$), hospital stay duration (Wilcoxon test, $p < 0.001$), and total duration of breastfeeding in the first year of life ($p = 0.014$), all being intrinsically related to delivery mode.

The number of children receiving exclusive formula feeding did not differ between the two groups. Only 36 children (30%) received antibiotics during their first year of life, some receiving multiple courses (56 courses in total for all children), mostly (80%) indicated for RIs.

**Microbiota composition and mode of delivery**. Of the 1243 fecal samples available from our 120 participants and their mothers, 1139 (92%) passed the quality criteria for further analysis following DNA extraction and 16S rRNA-based sequencing of the V4 hypervariable region (Supplementary Fig. 1), representing 70,886,595 high-quality reads in total. The Good's coverage of the included children's samples was high, with a minimum of 99.56% (median 99.96%). The raw Operational Taxonomical Unit (OTU)-table contained 623 OTUs distributed over seven bacterial phyla, with the Firmicutes generally being the most prominent phylum.

We observed that the infants' overall microbial community composition developed slowly toward an adult-like profile (mothers'), though had not yet reached full maturation to an adult-like composition at 12 months of age (Fig. 1a). This was illustrated by a steady decrease in Bray–Curtis (BC) dissimilarity index between infants' and mothers' samples over time, with a median index of 0.999 directly postpartum, which reached 0.739 at the end of the first year (Supplementary Table 1). We observed clear differences in the early development of the overall community composition between VD and CS children, with a maximum effect of delivery mode at 1 week of life (permutational multivariate analysis of variance [PERMANOVA] test, $R^2 = 0.142$, adjusted $p$-value 0.003, Benjamini–Hochberg method[11]) and significant differences until the age of 2 months ($R^2 = 0.021$, adjusted $p$-value 0.055), after which these differences gradually disappeared (Fig. 2). To rule out this finding was due to the indirect exposure of CS children to maternal antibiotics through breastfeeding, we repeated this analysis post hoc on a subset of 11 VD and 11 CS children who received exclusive formula feeding. We observed similar (at some timepoints even bigger) effects ($R^2$) of delivery mode on overall community composition until 2 months of life within this subset. The association between delivery mode and composition was still significant at 1 week (Fig. 2; Supplementary Table 2; $R^2 = 0.215$, adjusted $p$-value 0.008) and 2 weeks of life ($R^2 = 0.152$, adjusted $p$-value 0.044) in the exclusively formula fed children. For the whole study population, we also found that the microbial community in VD children was more stable when compared with CS children until 2 months of life (Fig. 1b). The BC dissimilarity between consecutive samples until 4 months was higher in CS, compared with VD children (Mann–Whitney test, $p < 0.001$, $p = 0.001$, and $p = 0.042$, for the intervals 1–2 weeks, 2 weeks–1 month and 1–2 months, respectively). In general, alpha diversity increased directly after birth, and again after 4 months of life, coinciding with the age that solid food was introduced to the children's diet (median = 128 days, IQR = 119.8–164.2 days). There were no significant differences in alpha diversity between the two delivery mode groups at any timepoint (Supplementary Fig. 2). When testing the effect of delivery mode on alpha diversity longitudinally with a mixed effect model, no significant effect was found (ANOVA, $p = 0.511$), and neither did feeding type have an effect in this model ($p = 0.652$).

Covariates that were significantly associated with fecal microbiota composition over time, as tested with the adonis2 function[12] (PERMANOVA test) were, besides mode of delivery ($R^2 = 0.013$, adjusted $p$-value $= 0.001$): age ($R^2 = 0.034$, adjusted $p$-value $= 0.001$), breastfeeding ($R^2 = 0.007$, adjusted $p$-value $= 0.001$), daycare attendance ($R^2 = 0.006$, adjusted $p$-value $= 0.001$), siblings

**Table 1 Baseline characteristics**

|  | Vaginal birth | C-section birth | p |
|---|---|---|---|
| n (%) | 74 (61.7) | 46 (38.3) |  |
| Gender, female (%) | 39 (52.7) | 24 (52.2) | 1.000 |
| Gravidity mothers, median (IQR) | 2.00 (1.00, 3.00) | 2.00 (2.00, 2.75) | 0.638 |
| Gestational age in weeks, mean (SD) | 39.75 (1.21) | 39.12 (0.84) | **0.003** |
| Birth weight in grams, mean (SD) | 3490.41 (485.87) | 3618.00 (459.18) | 0.156 |
| Ruptured membranes in hours, mean (SD) | 7.32 (9.82) | 3.05 (8.39) | **0.019** |
| Apgar score at 5 min (%) |  |  | 0.737 |
| 6 | 1 (1.4) | 0 (0.0) |  |
| 7 | 2 (2.7) | 1 (2.2) |  |
| 8 | 1 (1.4) | 2 (4.3) |  |
| 9 | 10 (13.7) | 8 (17.4) |  |
| 10 | 59 (80.8) | 35 (76.1) |  |
| Season of birth (%) |  |  | 0.557 |
| Winter | 13 (17.6) | 12 (26.1) |  |
| Spring | 17 (23.0) | 12 (26.1) |  |
| Summer | 29 (39.2) | 13 (28.3) |  |
| Fall | 15 (20.3) | 9 (19.6) |  |
| Hospital stay in dayparts, median (IQR) | 3.00 (1.00, 4.75) | 12.00 (9.50, 14.00) | **<0.001** |
| Number of siblings, median (IQR) | 1 (1.00, 1.00) | 1 (0.00, 1.00) | 0.264 |
| Presence of siblings < 5 years of age (%) | 40 (54.1) | 28 (60.9) | 0.587 |
| Presence of pets (%) |  |  | 0.910 |
| None | 41 (55.4) | 25 (54.3) |  |
| Cat(s) | 16 (21.6) | 11 (23.9) |  |
| Dog(s) | 6 (8.1) | 4 (8.7) |  |
| Cat(s) and dog(s) | 3 (4.1) | 3 (6.5) |  |
| Other | 8 (10.8) | 3 (6.5) |  |
| Inhouse smoking (%) | 1 (1.4) | 2 (4.3) | 0.674 |
| Parents finished higher education (%) | 60 (81.1) | 31 (67.4) | 0.138 |
| Breastfeeding in days, median (IQR) | 132.50 (7.00, 310.25) | 25.00 (1.00, 124.00) | **0.014** |
| Exclusive formula feeding (%) | 11 (14.9) | 11 (23.9) | 0.316 |
| Age start solid food in days, median (IQR) | 130.50 (118.25, 165.00) | 128.00 (120.00, 163.00) | 0.816 |
| Pacifier use at 1 month of age (%) | 53 (71.6) | 33 (71.7) | 1.000 |
| Antibiotic use in 1st year of life (%) | 19 (26.0) | 17 (37.8) | 0.254 |
| Number of antibiotic courses, median (IQR) | 0.00 (0.00, 0.75) | 0.00 (0.00, 1.00) | 0.119 |
| Daycare since (%) |  |  | 0.743 |
| 2 months | 1 (1.4) | 0 (0.0) |  |
| 3 months | 18 (24.3) | 11 (24.4) |  |
| 4 months | 14 (18.9) | 12 (26.7) |  |
| 6 months | 8 (10.8) | 7 (15.6) |  |
| 9 months | 9 (12.2) | 3 (6.7) |  |
| 12 months | 1 (1.4) | 0 (0.0) |  |
| >12 months | 23 (31.1) | 12 (26.7) |  |
| Fever, median (range) | 2.00 (0.00, 4.00) | 2.00 (0.00, 5.00) | 0.448 |
| Nausea postpartum, median (range) | 0.00 (0.00, 2.00) | 0.00 (0.00, 1.00) | 0.541 |
| Constipation, median (range) | 0.00 (0.00, 5.00) | 0.00 (0.00, 5.00) | 0.496 |
| Diarrhea, median (range) | 0.00 (0.00, 2.00) | 0.00 (0.00, 2.00) | 0.108 |
| Vomiting, median (range) | 0.00 (0.00, 3.00) | 0.00 (0.00, 1.00) | 0.505 |
| Thrush, median (range) | 0.00 (0.00, 4.00) | 0.00 (0.00, 3.00) | 0.316 |
| Respiratory tract infections (%) |  |  | 0.100 |
| 0–2 | 30 (41.1) | 11 (24.4) |  |
| 3–7 | 43 (58.9) | 34 (75.6) |  |

Baseline characteristics stratified by mode of delivery. Categorical variables are shown in absolute numbers with percentages (%); continuous, normally distributed variables as means with standard deviations (SD); continuous, non-normally distributed variables as medians with interquartile ranges (IQR) or ranges where specified. Two sample $t$ tests were used to compare means of normally distributed continuous variables; Wilcoxon rank-sum tests were applied to compare medians of non-normally distributed continuous variables; significant differences between categorical variables were tested with chi-square tests. The $p$-values of variables that differed significantly between the two groups are in bold and italicized for clarity. Source data are provided as a Source Data file

< 5 years of age ($R^2 = 0.006$, adjusted $p$-value $= 0.001$), pacifier use ($R^2 = 0.005$, adjusted $p$-value $= 0.003$) and antibiotics in the 4 weeks prior to sampling ($R^2 = 0.003$, adjusted $p$-value $= 0.03$). Pets in the household ($R^2 = 0.006$, adjusted $p$-value $= 0.061$) and duration of hospital stay after birth ($R^2 = 0.002$, adjusted $p$-value $= 0.061$) showed a trend toward being associated with fecal microbiota composition over time (Supplementary Fig. 3).

**Fecal microbiota seeding from mother to infant.** To study the existence of direct maternal fecal microbiota seeding during birth,

and to assess the role of delivery mode herein, we studied the concordance of the microbiota composition of children's fecal samples and their mother's microbiota over time, in relation to the concordance with other mothers' samples. Using a linear mixed model, we found that an infant's fecal microbiota composition was significantly more similar to that of its own mother than to that of other mothers in VD children when studied over the entire first year of life (ANOVA, $p = 0.025$; Fig. 3), but not in CS children ($p = 0.271$). This difference between groups seemed independent of the intravenous antibiotics administered to the mothers in the CS group after cord clamping, as the overall fecal

**a**

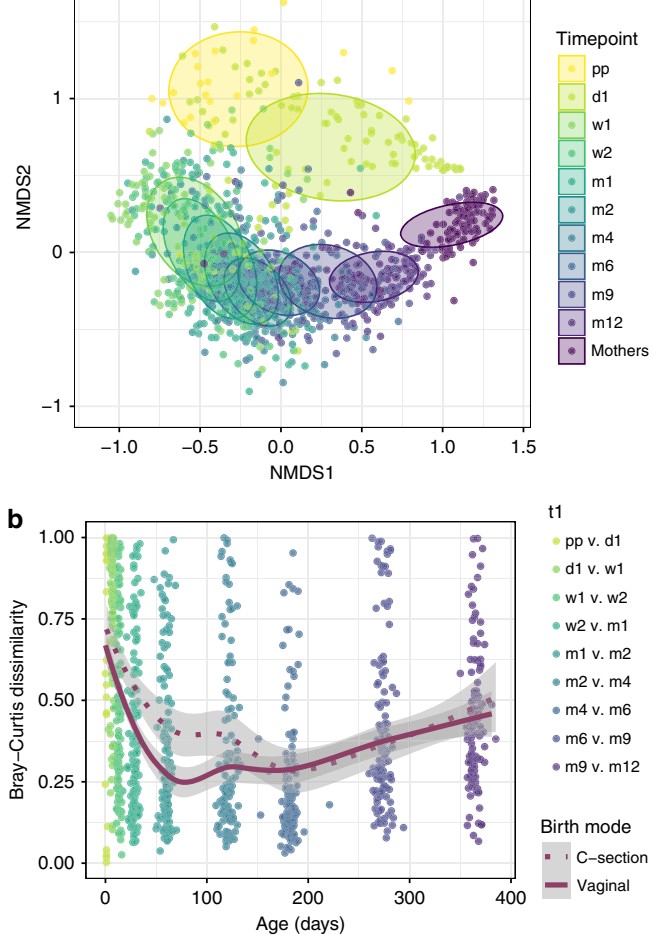

**b**

**Fig. 1** Overall gut microbiota community composition development and stability. **a** Nonmetric multidimensional scaling (nMDS) plot, based on Bray–Curtis (BC) dissimilarity between samples, with data points and ellipses colored by timepoint. Children's overall gut community composition developed toward a more adult-like pattern in the first year of life, becoming more similar to microbiota of adults (mothers' samples, n = 87). **b** As measure of stability, we calculated BC dissimilarities between consecutive sample pairs belonging to an individual per time interval and plotted these at the end of each interval (t + 1). Loess lines were fitted over the data points per delivery mode group, and the gray areas represent the 0.95 confidence intervals. Stability was significantly lower in C-section born infants until 2 months of life. Source data are provided as a Source Data file

microbiota composition of CS and VD mothers themselves did not differ shortly after birth (PERMANOVA test, $R^2 = 0.013$, $p = 0.351$; Supplementary Fig. 4).

**Dynamics of microbiota development**. The succession pattern of bacterial taxa in the VD children in our study population was consistent with the description of normal early-life gut microbiota development in previous studies (Fig. 4a)[13,14]. Facultative anaerobic genera, such as *Escherichia*, and *Staphylococcus* were highly abundant in the earliest samples, gradually making way for a predominance of the genus *Bifidobacterium*. Using smoothing spline analysis of variance (SS-ANOVA), we observed, among others, that *Bifidobacterium* was more abundant in VD than in CS children from day 1 until day 30, even when correcting for breastfeeding (adjusted p-value 0.003; Supplementary Table 3). Also, *Escherichia* was more abundant in VD compared with CS

born children in the first 85 days of life. In contrast, in CS children we found, among others, higher abundances of *Klebsiella* from birth to day 139 and *Enterococcus* between 7 and 35 days (both adjusted p-values 0.003). The dynamics of differences in *Bifidobacterium*, *Escherichia*, *Klebsiella*, and *Enterococcus* over time are visualized in Fig. 4b, underlining the effect size and duration of differences.

To confirm that these results were not the consequence of indirect antibiotic exposure of infants through breast milk, we executed a post hoc SS-ANOVA analysis for the subset of exclusively formula fed children. We analyzed the top five most abundant taxa over the first 2 months of life, and again found an increased abundance of *Bifidobacterium* (days 5–44, adjusted p-value = 0.003) in VD infants, and an increased abundance of *Klebsiella* in the CS born infants (days 10–20, adjusted p-value 0.020). Also, *Staphylococcus* was found to be more abundant in the CS children from 0 to 6 days (adjusted p-value = 0.020).

We used mixed effect models to study the potential associations between delivery mode, age, feeding type, antibiotic use and hospital stay duration, and the five most abundant taxa. We found that the abundance of *Bifidobacterium* was associated with mode of delivery (ANOVA, $p = 0.004$), age ($p < 0.001$), and breastfeeding ($p < 0.001$). Surprisingly, breastfeeding did not compensate for the lack of *Bifidobacterium* in children born by CS: children born by CS and receiving breastfeeding had less *Bifidobacterium* present in their fecal samples than formula fed VD children (at 1 week of life, Wilcoxon test, $p < 0.001$, median relative abundance 0.016 and 45.1%, respectively). The triad CS birth, age, and formula feeding were also positively associated with the abundance of *Enterococcus* and *Klebsiella*. In addition, *Klebsiella* abundance was positively associated with having received antibiotics in the previous 4 weeks of life ($p = 0.001$). *Escherichia* abundance was only associated with vaginal delivery ($p = 0.003$) and age ($p < 0.001$). *Staphylococcus* abundance was associated with age and breastfeeding (both $p < 0.001$), but not with delivery mode. We did not find an association between duration of hospital stay after birth and any of these taxa.

Among the remaining mode of delivery-associated taxa observed (see Supplementary Table 3), we found to be of particular interest that *Bacteroides* spp., which are considered to be important regulators of intestinal immunity[15], were more abundant in the VD compared with CS children in the first months of life.

**Delivery mode-induced microbiota changes and infant health**. Since delivery mode is reported to be associated with infant and childhood health, especially regarding respiratory illness[16], we defined a secondary research question, namely whether gut microbiota development is associated with health outcome. Although it was not our aim to study differences in health outcomes between the delivery mode groups in our cohort, we did find a trend toward differences in infectious disease and treatment parameters, specifically parent-reported RI events and antibiotic courses over the first year of life (Table 1, chi-square test, $p = 0.119$ and $p = 0.100$, respectively). Exploring this further with a temporal post hoc analysis, we additionally found a trend toward a lower hazard ratio for antibiotic prescriptions in VD children in the first year of life (Cox proportional hazard model, HR = 0.606, $p = 0.134$).

Altogether, these results supported the validity of our secondary aim to investigate the potential role of delivery mode-induced gut microbiota changes on health. To test this, we studied the association between fecal microbiota composition at 1 week of life (where the maximum effect of mode of delivery on microbiota composition was observed; Fig. 2), and all commonly observed

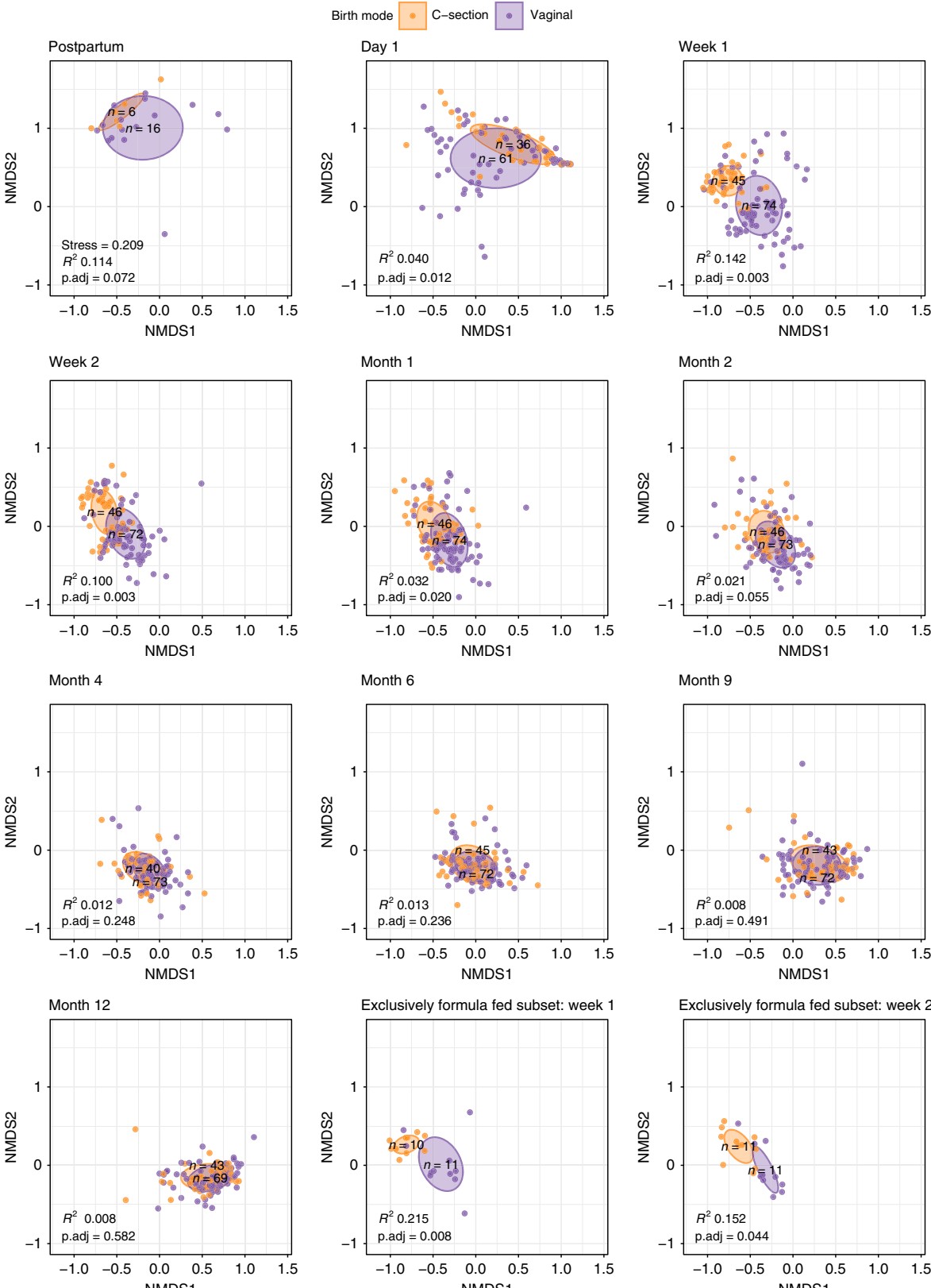

**Fig. 2** NMDS plots of children's samples per timepoint stratified according to mode of delivery. Nonmetric multidimensional scaling (nMDS) plots, based on Bray–Curtis (BC) dissimilarity between samples, visualizing the overall gut bacterial community composition stratified for mode of delivery, per timepoint. Each data point represents the microbial community composition of one sample. The ellipses represent the standard deviation of data points belonging to each birth mode group, with the center points of the ellipses calculated using the mean of the coordinates per group. The stress of the ordination, effect sizes ($R^2$) calculated by multivariate permutational multivariate analysis of variance (PERMANOVA) tests and corresponding adjusted p-values (p.adj) are shown in the plots, and n represents the biologically independent samples per group

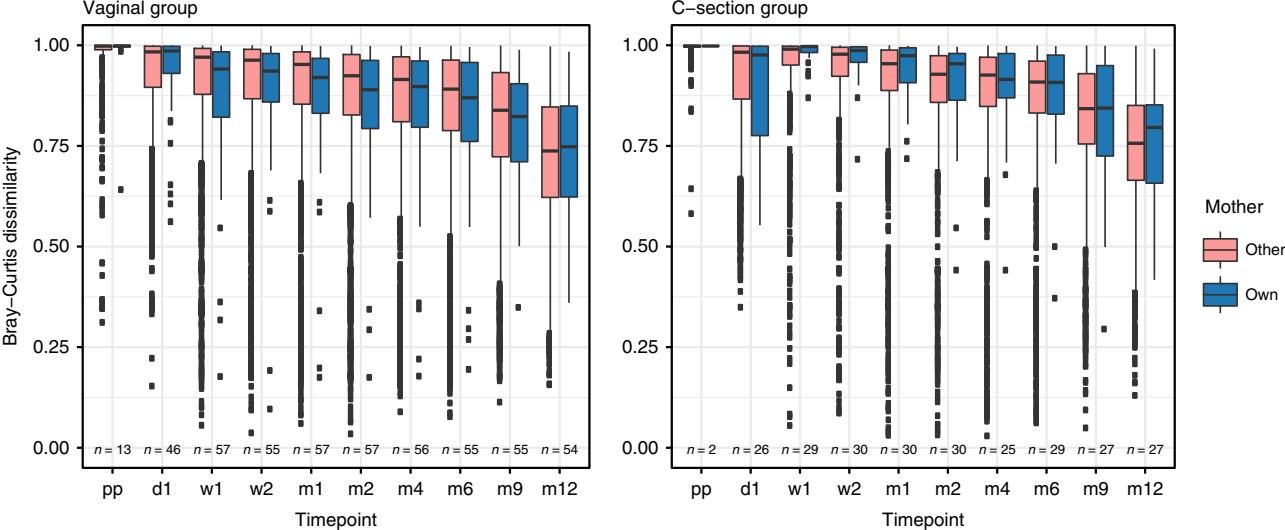

**Fig. 3** Comparison of overall composition between children and mothers (own vs. other). Children's fecal microbiota were compared to the mothers' fecal microbiota, based on BC dissimilarity and stratified according to mode of delivery. A significantly lower dissimilarity (more comparable microbiota) was observed between a child's microbiota and its own mother vs. other mothers in children born vaginally throughout the first year of life, but not in children born by C-section (linear mixed models, ANOVA; $p = 0.025$ and $p = 0.271$, respectively). Boxplots with medians are shown; the lower and upper hinges correspond to the first and third quartiles (the 25th and 75th percentiles); the upper and lower whiskers extend from the hinge to the largest and smallest value no further than 1.5*IQR from the hinge; outliers are plotted individually. Pp = postpartum, d = day, m = month, n = number of mother-own-infant pair comparison per timepoint. Source data are provided as a Source Data file

health parameters in the first year of life. We categorized the number of RI events into 0–2 vs. 3–7 RIs, based on previous studies of the respiratory microbiome within this same cohort[17,18]. In these studies, RIs were initially categorized into three groups based on the normal distribution of this variable. The 0–2 RIs group was found to have the most stable development of the nasopharyngeal microbiota when compared with children suffering from >2 RIs in the first year of life, and was defined as the healthy reference group. While there were no correlations between microbiota composition and GI complaints, we observed an association between microbiota composition at 1 week of life and the categorized number of RI events (0–2 vs. 3–7 RI events: PERMANOVA test, $R^2 = 0.033$, adjusted $p$-value $= 0.028$) as well as number of antibiotic courses prescribed over the first year ($R^2 = 0.024$, adjusted $p$-value $= 0.055$). These two outcomes were related, as the antibiotics prescribed were mostly indicated for RIs. We next aimed to identify the taxa explaining this association between microbiota composition at 1 week and fewer RI events later in life by cross-sectional differential abundance analysis, while adjusting for mode of delivery. We observed, among others, *Bifidobacterium* to be associated with fewer RI events (0–2 vs. 3–7 RI events, zero-inflated Gaussian mixture model, log2 fold change (log2FC) 2.118, adjusted $p$-value 0.049, Fig. 5), whereas *Klebsiella* and *Enterococcus* were negatively associated with fewer RI events (log2FC −3.242, adjusted $p$-value $= 0.007$ and log2FC −2.838, adjusted $p$-value $= 0.009$, respectively). Other taxa found to be negatively associated with fewer RI events encompassed genera such as *Veillonella* and *Staphylococcus*. Random forest analysis was used to verify these results and identified once again *Enterococcus*, *Bifidobacterium*, and *Klebsiella* as the most important taxa driving the prediction of the categorized RI events in the first year of life (Supplementary Table 4). Furthermore, a stratified analysis for the VD and CS groups separately, showed similar associations between gut microbiota composition at 1 week of life and number of RI events for the two groups (PERMANOVA test, $R^2$ 0.003 and 0.005, $p$-value 0.040 and 0.068, respectively). Taxa associated with the number of RI events were comparable between the overall and stratified analyses, although for VD children we now only found

significance for *Enterococcus* (zero-inflated Gaussian mixture model, log2FC −2.525, adjusted $p$-value 0.074), whereas for the CS children, *Bifidobacterium* (log2FC 2.805), *Klebsiella* (log2FC −6.991) and *Enterococcus* (log2FC; −4.283) were all significantly associated with number of RIs (adjusted $p$-values 0.055, 0.010, and 0.055, respectively).

**Validation of the results with metagenomics and targeted qPCR.** To validate our primary findings independently, we executed whole genome shotgun (WGS) sequencing on a subset of 20 randomly selected samples collected at 1 week of life from 10 VD and 10 CS born children. WGS sequencing yielded a total of 119 unique bacterial taxa. The relative abundances of the top 12 OTUs and species of both sequencing methods are represented in Supplementary Fig. 5, and show highly comparable profiles. The most abundant *Bifidobacterium* species in the WGS data set were *B. longum*, *B. breve*, and *B. adolescentis*. The combined relative abundances of these three species strongly correlated with the most abundant *Bifidobacterium* of the 16S rRNA data set (Pearson's r 0.95, adjusted $p$-value < 0.001). In this way, we could also correlate the *E. coli*, *Staphylococcus*, *Klebsiella*, and *E. faecium* OTU abundance of the 16S rRNA data set with high certainty to the *E. coli*, *S. epidermidis*, *K. oxytoca*, and *E. faecium* species abundance in the WGS data set (Supplementary Table 5).

We also used the WGS sequencing data to validate the differences found by 16S rRNA sequencing in overall gut microbiota composition between VD and CS born children at 1 week of life. The results from the ordination using the WGS sequencing data are shown in Supplementary Fig. 6. Again, we found a significant effect of delivery mode on the overall gut microbiota composition at 1 week of life using WGS sequencing data (PERMANOVA test, $R^2 = 0.125$, $p = 0.01$).

Next, we compared the microbiota profiles obtained by WGS sequencing between VD and CS children and observed that the combination of *B. breve*, *B. longum*, and *B. adolescentis* (median relative abundance 72.2% in the VD vs. 0.074% in the CS born children, Wilcoxon test, $p = 0.002$), *K. oxytoca* (<0.001 vs.

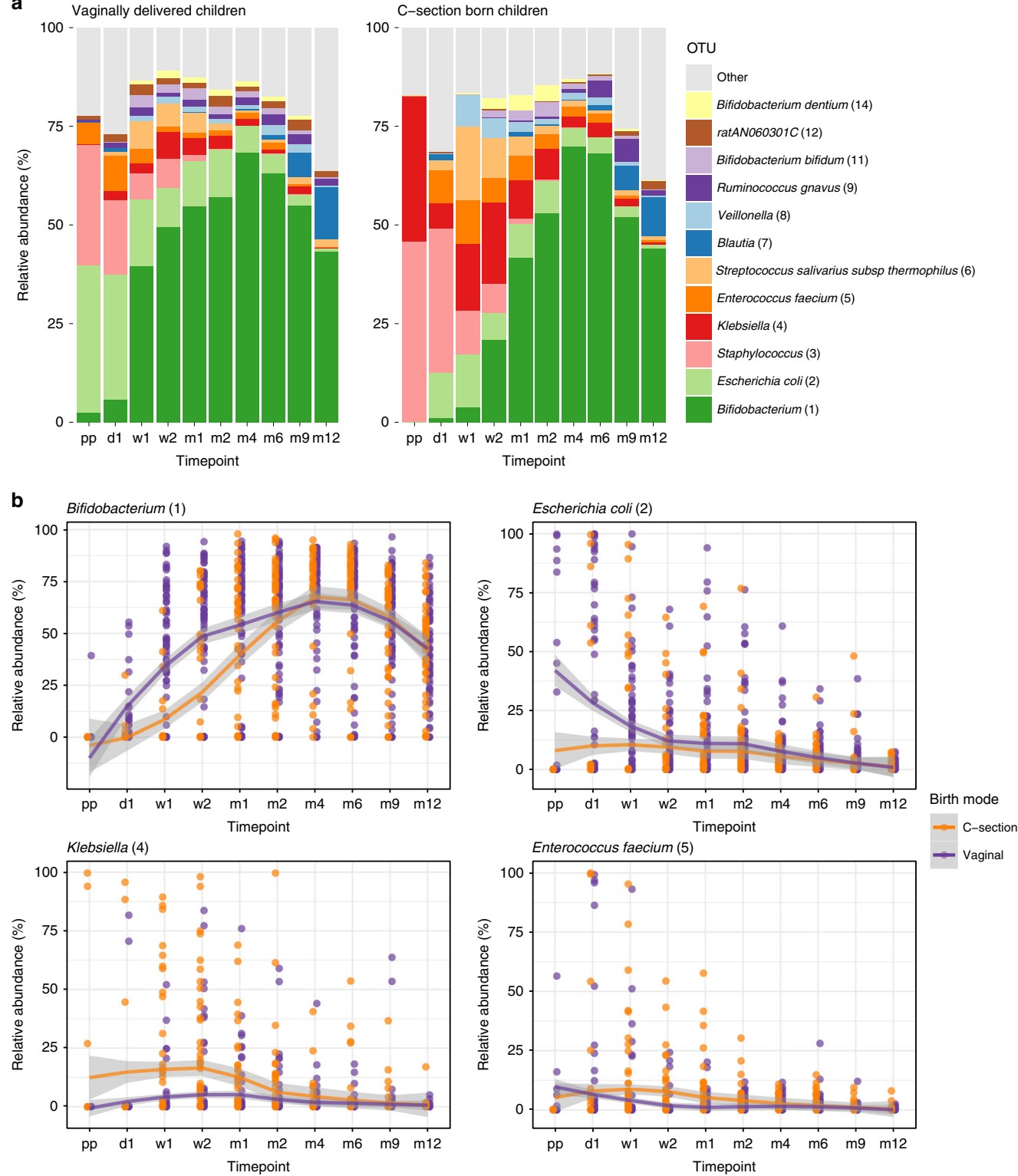

**Fig. 4** Mean relative abundance of most abundant OTUs. **a** Mean relative abundances of the 12 most abundant OTUs are depicted for all samples per timepoint, stratified by birth mode. Pp = postpartum, d = day, m = month. **b** Mean relative abundances of *Bifidobacterium*, *Escherichia*, *Klebsiella*, and *Enterococcus* over time. Loess lines were fitted over the data points per delivery mode group and the gray areas represent the 0.95 confidence intervals. Source data are provided as a Source Data file

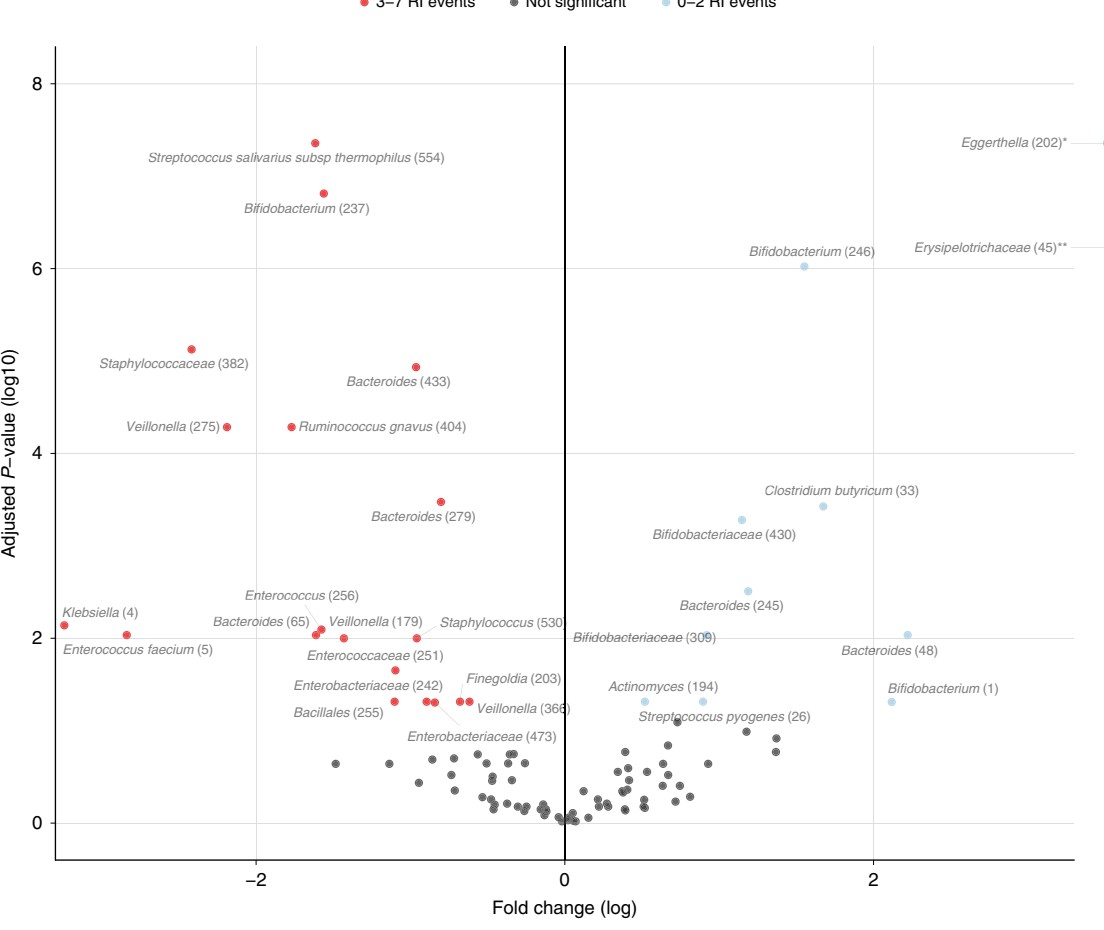

**Fig. 5** Differentially abundant taxa between 0-2 vs. 3-7 RI events in first year of life. To identify taxa that were differentially abundant between children experiencing more vs. limited respiratory infection (RI) events over the first year of life, fitZig analysis was performed on the 119 samples obtained at week 1 with the rare-features-filtered OTU-table containing 97 taxa and contrasts set to 0-2 vs. 3-7 RI events in the first year of life. The blue data points indicate taxa that were significantly more abundant in children having 0–2 RI events, while red points represent taxa that were significantly more abundant in children with 3–7 RI events in the first year of life. The results of two data points falling beyond the limits of the plot: *Eggerthella log2FC 3.512, adjusted p-value (log10) 7.357, **Erysipelotrichaceae log2FC 6.912, adjusted p-value (log10) 6.222, calculated using a zero-inflated Gaussian mixture model. Source data are provided as a Source Data file

0.006%, $p = 0.153$) and *E. faecium* (0.014 vs. 0.035%, $p = 0.023$) were differentially abundant between groups. In this data set, *E. coli* and *S. epidermidis* did not differ significantly between the delivery mode groups.

To alternatively validate our results on the overall cohort and in a targeted manner, we performed qPCR analyses for *E. coli*, *Klebsiella* spp. and *Enterococcus* spp. on all week 1 samples ($n = 119$). The qPCR results confirmed that *E. coli* is more commonly present in VD children compared with CS children (chi-square test, $p < 0.001$), whereas CS children are more often colonized with *Klebsiella* spp. ($p = 0.011$) and *Enterococcus* spp. ($p = 0.004$) than VD children, corroborating the 16S rRNA and WGS sequencing results (Supplementary Table 6). Finally, qPCR also confirmed that colonization with *Enterococcus* spp. and *Klebsiella* spp. at 1 week of life was positively associated with more RI events in the first year of life, though this difference was only significant for *Enterococcus* spp. ($p = 0.015$, Supplementary Table 7).

## Discussion

In this study, we were able to investigate the effect of mode of delivery on fecal microbiota development in healthy children, independent of maternal antibiotic exposure, as antibiotics given

perioperatively for CS were postponed to after cord clamping. We here describe the dynamics of the fecal microbiota in the first year of life in relation to mode of delivery and assess how early-life mode of delivery-induced microbiota alterations might affect susceptibility to RIs in the first year of life.

We found substantial differences in the gut microbiota composition and stability between VD and CS children, especially in the first months of life, with notably *Bifidobacterium* being more abundant in VD children, consistent with literature[19–23]. Bifidobacteria are health-associated microbes well-known for their use as probiotics[24]. They promote gut health and provide defense against pathogens[1]. We found that in children born by CS, the colonization with *Bifidobacterium* was significantly delayed, which was not affected by feeding type. This suggests that maternal transmission during vaginal delivery is essential in acquiring these bacterial species in early life[4], which was supported by the evidence we found for fecal seeding from mother to child in the VD, but not in CS children. Perhaps therefore not solely vaginal microbiota seeding[25] but also fecal microbiota seeding during vaginal delivery is instrumental in shaping the newborn's gut microbial environment. These data suggest that only after proper initial (vaginal-)fecal seeding takes place, the growth of beneficial groups such as *Bifidobacterium* can be promoted, which can be further enhanced through the prebiotic

oligosaccharides present in breast milk[26]. Hence, the stimulation of breastfeeding in women that have delivered by CS, or the advances in prebiotic formulations of modern formula milk, might not be able to correct the lack of *Bifidobacterium* seeding during delivery, as we saw that breastfed CS infants carried these bacteria in lower abundance than formula fed VD infants. Our data also suggest that this lack of *Bifidobacterium* is not the consequence of antibiotic exposure intrapartum, but merely a consequence of delivery mode itself, which was further supported by the sub-analysis on formula fed infants, limiting the likelihood of exposure to maternal antibiotics through breast milk. Importantly, the delay in *Bifidobacterium* establishment may have a major impact on the infants' early life and future health, as the window of opportunity for immune priming occurs within the first 100 days of life[27].

While *Bifidobacterium* was abundantly present in the fecal samples of VD children, the potential pathogenic and proinflammatory *Klebsiella* and *Enterococcus* were more abundant in children born by CS, which is in accordance with previous studies[22,28]. For *Klebsiella*, the difference in abundance lasted for more than 4 months, long after the initial neonatal period. Recently, an increased *Klebsiella*/*Bifidobacterium* ratio in early life was correlated with later development of pediatric allergy[29]. This could be an important link between mode of delivery and increased prevalence of pediatric allergies following CS birth. In addition, bacteria from the *Klebsiella* genus are a common cause of nosocomial infections and act as a reservoir for a diverse scale of antimicrobial resistance genes[30,31]. It is highly likely that the *Klebsiella* bacteria found in our study population were acquired from the hospital (operating room) environment, and in the absence of a stable environment, thrived in the gut of CS infants. These findings might thus imply that mode of delivery not only increases the risk for immunological disorders, but also the risk for (antibiotic resistant) infections.

Although our study was not powered to investigate differences in overall health outcomes between VD and CS children, we still found a trend toward more RI events and a higher need for antibiotics in the first year of life in CS compared with VD children. This early timeframe is clinically important, as early onset of RI is considered a risk factor for recurrent infections[32,33]. As CS incidence is specifically rising in Latin American countries[7], in future studies it would be of interest to assess socioeconomic factors as a link between mode of delivery, maternal microbiota characteristics, choice of feeding type and early-life risk of infection.

Uniquely to our study, we were able to relate the fecal microbiota composition at a very early age (one week), where the microbiota differences between mode of delivery groups were largest, to RI events occurring in the first year of life. Our stratified analyses per delivery mode group showed similar results to the overall cohort analysis, with especially *Bifidobacterium*, *Klebsiella*, and *Enterococcus* being associated with RI events independent of mode of delivery, though effect sizes were larger in the CS group. One reason for this could be a mediating effect of breastfeeding with mode of delivery, since breastfeeding is less common in CS children, and has a known protective and independent effect against infectious diseases[34].

Although these results require validation in preferentially larger cohorts, our findings do suggest that early-life gut microbiota composition might play a role in systemic (immune-mediated) resistance against infectious diseases. We found that especially early-life presence and abundance of *Klebsiella* and *Enterococcus* species, belonging to the ESKAPE pathogen family[35], show a relation with the development of a higher incidence of RI events later in life, whereas *Bifidobacterium* and certain *Bacteroides* species might play a protective role. Potential mechanisms by

which bifidobacteria may protect against pathogens and resulting infections is through increasing the local pH and indirectly through increasing the short-chain fatty acid abundance in the gut, promoting gut health[1].

Strengths of our study include high sampling frequency, the broad scale of clinical and epidemiological data collected consistently by trained research nurses, the quality of the sequencing data enabling us to maintain a strict filtering threshold, and the longitudinal character of our study combined with the use of differential abundance testing, providing us with enough statistical power to discern differences in both microbial succession patterns and disease parameters between delivery mode groups.

The most important limitation of our study is the limited number of samples from the postpartum timepoint that elicited sufficient amounts of DNA for characterization of the microbiota, though, inherent to the low dense colonization in neonates in general, this is unlikely to have introduced confounding. Also, sequencing of the V4 hypervariable region of the 16S rRNA gene does not allow for confident reporting of results on a lower taxonomical level than genus level, though this was in part resolved by WGS sequencing validation of a subset of 20 samples. Finally, our observational study was not primarily designed to investigate health differences between delivery mode groups, therefore the power to research correlations between drivers, biomarkers, and health consequences was limited.

In conclusion, we here report on modest differences in health characteristics between delivery mode groups, where children born by CS show a tendency toward higher incidence of RI events in early life, as well as a trend in higher need for antibiotics than VD children, with the former being linked to differences in abundance of several biomarker bacteria. In CS delivered children, the gut microbiota appears less stable with the acquisition of *Bifidobacterium* being delayed when compared with VD children. This delay is independent of feeding type, suggesting that maternal transmission during vaginal delivery is essential in acquiring these bacterial species in early life, which is further supported by the evidence for fecal seeding in the VD children, but not in CS children in our study. The abundance of potential pathogens from the genera *Klebsiella* and *Enterococcus* is higher in children born by CS, and independent of prenatal antibiotic exposure, duration of hospitalization, and feeding type. These taxa are also associated with a higher incidence of RI events in the first year of life. These findings provide evidence for a possible link between mode of delivery-induced alterations in the infant gut microbiota and susceptibility to (infectious) diseases.

## Methods

**Study population and sample collection**. Data and fecal samples were available from 120 children, of which 74 were vaginally delivered (VD) and 46 born by caesarean section (CS), and their mothers, who had participated in the prospective Microbiome Utrecht Infant Study (MUIS) consisting of healthy, full term Dutch children[36]. These 120 children had all (1) completed the 1-year follow-up (except for two children, for whom samples were only collected until 6 months, after which they dropped out due to moving out of the area) and (2) had at least five fecal samples available from ten timepoints. Recruitment took place during pregnancy and written informed consent was obtained from both parents. Ethical approval was granted by the national ethics committee in the Netherlands, METC Noord-Holland (committee on research involving human subjects, M012-015, NTR3986). The study was conducted in accordance with the European Statements for Good Clinical Practice. The study had originally been powered based on the abundance and distribution of previously published microbiota data from infants[37], ensuring a power of 0.8 to detect at least significant differences in alpha and beta diversity between groups, as well as differences in abundance of the 25 most important operational taxonomical units (OTUs), taking into consideration OTUs with high and low variability and abundance and varying effect sizes. This power calculation was later verified by an online (Human Microbiome Project based) tool[38]. We initially aimed to enroll 88 infants, 44 infants per delivery mode group, allowing a dropout of 10%. Due to uneven enrollment in both arms, approval was granted by the ethical committee to prolong enrollment to ensure sufficient CS recruitment, simultaneously continuing the parallel enrollment of VD infants to prevent

seasonal/annual differences in microbiota development between the delivery mode groups. Eventually, 78 VD and 52 CS children were recruited; 10 (7.7%) dropped out after an average of 2 weeks of follow-up.

Hospital and home visits took place within 2 h postpartum (pp), 24–36 h after delivery (d1), at 7 (w1) and 14 (w2) days and at 1, 2, 4, 6, 9, and 12 months (m) of age. Fecal samples were collected by a nurse during hospitalization or by the parents prior to the home visits. On a voluntary basis, mothers provided one fecal sample 2 weeks after childbirth. Sterile fecal containers were used for sample collection, and parents were instructed to store the material directly at −20 °C in the (home) freezer. Samples were transported on dry ice and transferred for long-term storage at −80 °C until further laboratory processing. Extensive questionnaires on the health status of the children were collected at each visit by research personnel and additionally at 3 months. At baseline, information was collected on prenatal and perinatal characteristics.

Respiratory infection (RI) events were defined as occurrence of fever (>38.0°) in combination with any of the following parent-reported symptoms: cough, wheezing, dyspnea, earache, or malaise. Gastrointestinal (GI) complaints were categorized into: postpartum nausea, constipation, diarrhea, and vomiting and were noted as being either present or absent since the previous home visit. For each symptom, occurrences over the first year were summed up to a cumulative number.

**DNA isolation and sequencing.** Fecal samples were thawed and homogenized by vortexing. For bacterial DNA extraction ~100 μl of raw feces was added to 300 μl of lysis buffer (Agowa Mag Mini DNA Isolation Kit, LGC ltd, UK), 500 μl of 0.1-mm zirconium beads (BioSpec products, Bartlesville, OK, USA) and 500 μl of phenol saturated with Tris-HCl (pH 8.0; Carl Roth, GMBH, Germany) in a 96-wells plate. The samples were mechanically disrupted using a Mini-BeadBeater-96 (BioSpec products, Bartlesville, OK, USA) for 2 min at 2100 oscillations per minute. DNA purification was performed using the Agowa Mag Mini DNA Isolation Kit according to the manufacturer's recommendations. The extracted DNA was eluted in a final volume of 60 μl of elution buffer (LGC Genomics, Germany). Samples collected directly postpartum and on day 1 were presumed to have low bacterial abundance and diversity. Therefore, adaptations in the standard DNA isolation procedure were applied: 150 μl of raw feces was added to 350 μl of lysis buffer, and bead beating was done for 2*3 min. In total, 300 μl of the aqueous layer was used for extraction with the Agowa Mag Mini DNA Isolation Kit, the binding time of DNA to the magnetic beads was prolonged to 30 min, the magnetic beads were washed twice with wash buffer 1, and the extracted DNA was eluted in a final volume of 40 μl of buffer. As the yield of these early samples was low, another round of DNA isolation was performed on the aliquots of this set with an altered protocol: a small loop of feces (~150 μl, or 100 μl when liquid) was added to 650 μl lysis buffer including zirconium beads and 600 μl of phenol, and the whole aqueous layer was used. DNA blanks and feces pools consisting of a mix of up to three random samples served as controls. The amount of bacterial DNA was determined by quantitative polymerase chain reaction (qPCR) as described[39], using primers specific for the bacterial 16S rRNA gene (forward: CGAAAGCGTGGGGAGCA AA; reverse: GTTCGTACTCCCCAGGCGG; Probe: 6FAM-ATTAGATACCCT GGTAGTCCA-MGB) on the 7500 Fast Real Time system (Applied Biosystems, CA, USA).

For the sequencing of the V4 hypervariable region of the 16S rRNA gene, 1 ng of DNA was amplified using F515/R806 primers and 30 amplification cycles[40,41]. After amplification of the V4 hypervariable region of the 16S rRNA, the amount of amplified DNA per sample was quantified with the dsDNA 910 Reagent Kit on the Fragment Analyzer (Advanced Analytical, IA, USA). Samples that yielded insufficient DNA after amplification, defined as <0.5 ng per μl, were repeated with a higher concentration of template DNA. Each PCR plate included a mock control and three PCR blanks. 16S rRNA sequencing was performed on the Illumina MiSeq platform (Illumina, Eindhoven, the Netherlands) on a total of 1139 samples and 85 controls in 17 runs.

**Bioinformatic processing.** The sequences were processed in our bioinformatics pipeline, where we applied an adaptive, window-based trimming algorithm (Sickle, version 1.33) to filter out low quality reads maintaining a Phred score threshold of 30 and a length threshold of 150 nucleotides[42]. Error correction was conducted with BayesHammer (SPAdes genome assembler toolkit, version 3.5.0)[43]. Each set of paired-end sequence reads was assembled with PANDAseq (version 2.10) and demultiplexed (QIIME, version 1.9.1)[44,45]. Singleton and chimeric reads (UCHIME) were removed. OTU picking was performed with VSEARCH abundance-based greedy clustering with a 97% identity threshold[46]. OTUs were annotated with the Naïve Bayesian RDP classifier (version 2.2) and the SILVA reference database[47,48]. This resulted in an OTU-table containing 6690 taxa. We made an abundance-filtered data set selecting OTUs present at a confident level of detection (0.1% relative abundance) in at least two samples[49], henceforth referred to as our raw OTU-table. The raw OTU-table consisted of 623 taxa (0.4% sequences excluded with filtering) and was used for the downstream analyses unless otherwise specified.

**Whole genome shotgun (WGS) sequencing and processing.** To validate the taxonomical annotation of the 16S rRNA sequences and some of our 16S rRNA-based findings, we performed whole genome shotgun (WGS) sequencing on a randomly selected subset of 20 fecal samples collected from 10 VD and 10 CS born children at the age of 1 week. Samples were prepared using the Truseq Nano gel free library preparation kit. Using a NovaSeq instrument, 150 base paired-end sequence data were generated from the libraries to yield 750 M + 750 M reads (two runs). Reads were trimmed using Cutadapt[50] (version cutadapt-1.9.dev2) of amplicon adapter sequences and on quality at the 3′ end maintaining a quality threshold of 30 and a minimum read length of 35 base pairs. Per sample and per-run SAM files were generated with Bowtie2[51] and the MetaPhlAn2[52] database while adhering to recommended parameters and using -no-unal to suppress reporting unaligned reads and the -very-sensitive parameter. Each SAM file was assigned a read group and SAM files from different runs were merged sample-wise using Picard[53]. MetaPhlAn2 was run to identify the bacterial taxa present within each sample. The SAM files generated using Bowtie2 were used as input for MetaPhlAn2, all other parameters were kept as default.

**Determination of specific biomarkers by qPCR.** To identify the presence of *E. coli, Klebsiella* spp. (including *K. oxytoca* and *K. pneumoniae*) and *Enterococcus* spp. in all the samples collected at 1 week of age (n = 119), we used the VetMAX™ MastiType Multi Kit (Applied Biosystems™, CA, USA) according to the manufacturer's instructions. The qPCR test results were analyzed with the recommended Animal Health VeriVet Software, available on Thermo Fisher Cloud. One sample was discarded from the statistical analyses because its Internal Amplification Control did not pass the Ct-value criteria in three out of the four mixes.

**Statistical analyses.** A statistical analysis scheme showing the flow in and order of analyses to address the primary, secondary and exploratory research questions can be found in Supplementary Fig. 7. All analyses were performed in R version 3.4.3[54] within RStudio version 1.1.383[55], and figures were made using packages ggplot2[56] and ggpubr[57]. For simple, independent comparisons, we considered p-values <0.05 to be significant. However, for all analyses regarding multiple comparisons, we used the Benjamini–Hochberg method to correct for multiple testing[11].

For comparisons of group differences, a two-sample t test, Wilcoxon rank-sum test or chi-square test was used where appropriate. Survival analysis was executed using the packages *survival* and *survminer*[58,59]. The hazard ratio for antibiotic administration was calculated with a Cox proportional hazard model.

Group differences in Shannon alpha diversity were calculated with t-tests and a linear mixed-effect model with participant as random effect while correcting for age and feeding type. Differences in overall gut bacterial community composition were visualized with nonmetric multidimensional scaling plots (nMDS; *vegan* package[12]). Ordinations were based on the BC dissimilarity matrix of relative abundance data with parameter trymax 10,000. The overall gut bacterial community composition of children born by emergency CS (n = 10) was, although not fully similar, more similar to that of children born by planned CS (Supplementary Fig. 8; permutational multivariate analysis of variance (PERMANOVA) test, $R^2 = 0.005$, $p = 0.051$) than by vaginal birth ($R^2 = 0.006$, $p = 0.002$). Because the number of children born by emergency CS was too small to analyze as independent group, we decided to group emergency and planned CS children together for all delivery mode comparisons.

To study whether the differences in overall gut microbiota community composition between delivery mode groups were not influenced by antibiotics received indirectly through the breast milk from mothers treated prophylactically after CS, post hoc analyses were performed on a subset of 11 VD and 11 CS infants that received exclusive formula feeding.

Associations between clinical outcome and microbiota composition were analyzed with the adonis2 function (*vegan* package[12]), based on PERMANOVA tests per timepoint and across all timepoints using 1999 permutations, including all variables that showed a significant association with microbiota composition when tested individually, namely mode of delivery, duration of hospital stay after delivery, breastfeeding at time of sampling, pacifier use, antibiotics in the 4 weeks prior to sampling, siblings <5 years in the household, pets in the household, high education of parents, and daycare attendance. For the temporal analyses, age and subject were added to control for repeated measures.

To test the occurrence of fecal seeding (i.e., whether the microbiota composition in VD children was more similar to that of their own mothers than in the CS group), BC dissimilarities were calculated between children's and their own mother's vs. other mothers' samples, stratified per group. We used a linear mixed model to assess the effect of delivery mode (fixed effect) on BC dissimilarity as a dependent variable, while including subject as a random intercept to adequately control for repeated measures and correcting for timepoint, using the lme function of the *nlme* package[60]. P-values of our linear mixed models were extracted using the ANOVA function.

The stability of the gut microbiota composition in the first year of life was visualized by measuring the BC dissimilarities between consecutive samples within each participant over time. To test for group differences, the Mann–Whitney test was used.

Individual bacterial taxa and their succession patterns, and potential differences thereof between groups, were studied at the lowest taxonomic annotated level (OTU). Differential abundance testing was executed with smoothing spline analysis of variance (SS-ANOVA, fitTimeSeries function, *metagenomeseq* package[61,62])

allowing not only to detect biomarker OTUs related to mode of delivery, but also to identify the specific intervals in which significant differences existed. For this analysis, the raw OTU-table was filtered and OTUs with >10 reads in ≥50 samples were included, resulting in 306 OTUs (of the 623)[17]. The SS-ANOVA analysis was adjusted for covariates that both had (1) an effect on overall gut microbiota composition and (2) were unevenly distributed between the two delivery mode groups, namely hospital stay duration after birth, breastfeeding at time of sampling and antibiotic use in the 4 weeks prior to sampling. Although duration of ruptured membranes and gestational age were associated with mode of delivery, we did not find an association with gut microbiota composition, therefore these two variables were excluded from downstream fitTimeSeries analyses. A post hoc SS-ANOVA was performed on the subset of children who were exclusively formula fed, specifically testing for differences in the top five most abundant taxa and focusing on the first 2 months of life, as differences between delivery mode groups overall were most pronounced in this period.

We used linear mixed models to test the importance of delivery mode, duration of hospital stay after birth, feeding type, and antibiotic use on the relative abundance of the top five most abundant taxa. Using the lme function[60], we set the clinical variables as fixed effects and the arcsine square root transformed relative abundances of each taxon of interest as dependent variable, adding subject as random intercept and correcting for age.

To test if the microbiota could predict for cumulative disease parameters (namely fever episodes, thrush, GI symptoms, RI events, general practitioner and specialist consultation, and antibiotic prescription) after 1 week of age, we performed a BC-based PERMANOVA on the overall community composition of the samples obtained at this sampling moment. The number of RI events in the first year of life was tested as both a continuous and categorical variable grouped in 0–2 and 3–7 episodes, based on previous studies[17,18]. The fitZig function of the metagenomeSeq package was used to assess the driving OTUs behind significant predictions[62], after removing rare features present in <10 samples, resulting 97 OTUs included in this analysis. The analysis was adjusted for delivery mode. Random forest analysis was used to verify these results, setting the categorized number of RI events as outcome, and the OTUs present in the samples at 1 week of age as predictors along with delivery mode and variables also adjusted for in the fitTimeSeries analysis[63]. We also performed the fitZig analysis in a stratified manner for both delivery mode groups.

The results from 16S rRNA sequencing at 1 week of life were validated by untargeted WGS sequencing (subset of 20 infants) and targeted qPCR (on all 120 infants).

**Reporting summary**. Further information on research design is available in the Nature Research Reporting Summary linked to this article.

## Data availability

Sequence data that support the findings of this study have been deposited in the NCBI Sequence Read Archive (SRA) database with BioProject ID PRJNA481243 and PRJNA555020. The source data underlying Table 1, Fig. 1a, b, 2, 3, 4a, b and 5, Supplementary Tables 1 and 5–7 and Supplementary Figs. 2, 5, and 6 are provided as a Source Data file.

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

## Acknowledgements

The authors are most grateful for the participation of all the children and their families. We would like to acknowledge all the members of the research team of the Spaarne Gasthuis Academy, the Department of Obstetrics and Gynecology of the Spaarne Gasthuis (Hoofddorp and Haarlem, the Netherlands) and participating midwifery clinics. We would also like to acknowledge Edinburgh Genomics for executing the WGS sequencing and Katherine Emelianova for the bioinformatic analysis of the shotgun sequences. This work was supported in part by the Netherlands Organisation for Scientific Research (NWO-VIDI; grant 91715359) and CSO/NRS Scottish Senior Clinical Fellowship award (SCAF/16/03). The study was co-sponsored by the Spaarne Hospital Hoofdorp and Haarlem and the University Medical Centre Utrecht, The Netherlands.

## Author contributions

M.A.v.H., E.A.M.S. and D.B. conceived and designed the experiments. M.A.v.H. and A.A. T.M.B. included the participants. M.L.J.N.C., K.A. and R.L.W. were responsible for the execution and quality control of the laboratory work. M.R., W.M., S.F. and D.B. analyzed the data. M.R., M.A.v.H., D.vB., E.A.M.S., S.F. and D.B. wrote the paper. All authors significantly contributed to interpreting the results, critically revised the manuscript for important intellectual content, and approved the final paper.

## Competing interests

E.A.M.S. declares to have received unrestricted research support from Pfizer, grant support for vaccine studies from Pfizer and GSK, and fees paid to the institution for advisory boards or participation in independent data-monitoring committees for Pfizer and GSK. D.B. declares to have received unrestricted fees paid to the institution for advisory work for Friesland Campina and well as research support from Nutricia. None of the fees or grants listed here were received for the research described in this paper. No other authors reported financial disclosures. None of the other authors report competing interests.

## Additional information

**Peer Review Information** *Nature Communications* thanks Michael Grayling, Geraint Rogers and the other, anonymous, reviewer for their contribution to the peer review of this work. Peer reviewer reports are available.

