## [Peer Review File · Nature Communications]

Reviewers' comments:

Reviewer #1 (Remarks to the Author):

I wish to make it clear that I have reviewed a version of the manuscript previously (Reviewer 1, as per rebuttal). I would like to congratulate the authors on a very thorough and well-considered rebuttal. I think their manuscript has been much improved by the revisions made (I hope they found the comments helpful).

Line 132 – I think an apostrophe should be to “mothers”, although unclear whether this was a pair- or group-wise comparison

Geraint Rogers

Reviewer #2 (Remarks to the Author):

I still struggle with the authors' interpretation of the health effects of CS versus VD infants. I agree that the study was underpowered to investigate These effects, yet the authors still state that "...children born by CS show higher incidence of RI events in early life, as well as a higher need for antibiotics..." (line 380). The finding that Enterococcus was associated with RI Events in both the VD and CS Groups clearly argues against a role of an adverse health of Enterococcus by mode of delivery. Rather the microbial composition, in particular Enterococcus is associated with RI independently of mode of delivery.

In line 310-311 the authors "suggest that maternal transmission during vaginal delivery is essential in acquiring These bacterial species". This statement should mention 'in early life' since is really is the early but not alte acquisition which is absolutely comparable between groups after month 1 (figure 3B). I feel that much of this part of the discussion is highly speculative as no clear health effects can be demonstrated in this study but a non-significant trend given the limited number of included subjects.

Reviewer #4 (Remarks to the Author):

Title: Impact of delivery mode-associated gut microbiota dynamics on health in the first year of life

Author(s): Reyman M, van Houten MA, van Baarle D et al

Manuscript ID: NCOMMS-19-11133-T

Date review requested: May 14 2019

Date Review Submitted: May 30 2019

Reviewer: Michael Grayling, Newcastle University

0. Summary

I concur with the sentiment of Reviewer 1 that this paper discusses a very interesting and important issue. However, I also agree with Reviewer 2 that it is a little alarming that no analysis protocol (of any form) was pre-specified by the authors, and that an extremely large number of analyses have been performed.

Nonetheless, I sympathise with the authors that this is not regular practice in their field, and that it may also be difficult to achieve. Furthermore, I found the newly provided flow diagrams on the performed analyses to be very clear, and I feel that the authors have addressed the previous reviewers' criticisms effectively. My comments therefore are mostly minor, seeking to address a small number of additional issues that I identified.

1. Major comments

1.1. No justification appears to be provided by the authors as to why the number of RI events was dichotomised in to 0-2 and 3-7. This is troubling as it is easy to envisage that the presented results (which are discussed at length in the manuscript) may not be present if 0-3 and 4-7 was chosen instead. How was this categorisation chosen?

2. Minor comments

2.1. Clarification should be added to the online supplementary methods on how p-values were calculated for analyses performed using linear mixed models. I presume if nlme was used, that these are the default returned p-values that assume a denominator DoF equal to a corresponding balanced ANOVA design? (c.f., lme4 no longer returns p-values – this is not in general a simple issue.)

2.2. There are several ways in which I feel the presented figures could be improved:

- Where at all possible, colour-blind friendly palettes should be used.
- In the legends of figures that contain ellipses, clarification on the reason for the location of the centre point should be added (not just detail that the ellipse shape is dependent on standard deviations).
- Superscripts and subscripts are now easy to incorporate, so text such as "R²" should be corrected.
- In Figure 4, there are labels for which no point can be seen, and labels that are obscured behind points; this should be resolvable by tweaking parameters in ggrepel.
- In Supplementary Figure 2, consider using jittering and reducing the length of the horizontal portion of the error bars in order to limit overlay.

2.3. Lines 114-115; I would clarify whether the cases with pre-/intrapartum antibiotic use were included in the analyses.

2.4. Lines 131-133; I would add the formal statistical justification to back up the statement that the infant microbial community composition developed towards an adult-like profile.

3. Grammatical/typographical

The following may be helpful:

3.1. L202 "we would" not "we'd"

3.2. HR on L213, should this be 0.501 not -0.501?

3.3. L275; RIs not respiratory infections.

Reviewer #1

I wish to make it clear that I have reviewed a version of the manuscript previously (Reviewer 1, as per rebuttal). I would like to congratulate the authors on a very thorough and well-considered rebuttal. I think their manuscript has been much improved by the revisions made (I hope they found the comments helpful).

We thank the Reviewer for the positive comments and congratulations. Indeed, we found the comments extremely helpful and agree that by answering the critical questions our manuscript has been much improved.

Specific comments

Line 132: I think an apostrophe should be added to “mothers”, although unclear whether this was a pair- or group-wise comparison.

We thank the Reviewer for this suggestion. Adding an apostrophe here will indeed help clarify that over time, the composition of the children’s samples became more similar to the composition of the mothers’ samples. We have adjusted the text accordingly at page 10 line 143. Additionally, we have statistically justified our statement as per Reviewer 4’s suggestion (see below).

Reviewer #2

Specific comments

I still struggle with the authors' interpretation of the health effects of CS versus VD infants. I agree that the study was underpowered to investigate these effects, yet the authors still state that "...children born by CS show higher incidence of RI events in early life, as well as a higher need for antibiotics..." (line 380).

We thank the Reviewer for the constructive comments. We regret that although we toned down the results regarding the health effects throughout our manuscript, now reporting on trends in differences in respiratory infection (RI) events and antibiotic prescriptions between the caesarean section (CS) and vaginally delivered (VD) infants, we mistakenly overlooked this sentence in our discussion. While we acknowledge that our study was underpowered to investigate health effects, we do find a trend towards higher RI events and antibiotic prescriptions in the CS children and believe it is of enough clinical relevance to report this. We have now adjusted our statement at page 20, lines 365-368 in our revised

manuscript as follows: "...children born by CS show a tendency towards higher incidence of RI events in early life, as well as a trend in higher antibiotic prescriptions...".

The finding that Enterococcus was associated with RI events in both the VD and CS groups clearly argues against a role of an adverse health of Enterococcus by mode of delivery. Rather the microbial composition, in particular Enterococcus is associated with RI independently of mode of delivery.

We agree fully with the Reviewer that *Enterococcus* abundance is associated with RIs, independently of mode of delivery and have now added this on line 339 at page 18. However, we also found *Enterococcus* to be strongly enriched in children born by CS, thereby possibly making them more susceptible for RI events. Altogether, this is why we propose a possible link between delivery mode-induced changes in the gut microbiota and infant health as we conclude at page 20, lines 372-377 of our revised manuscript. We hope the Reviewer is satisfied with this more nuanced discussion.

In line 310-311 the authors "suggest that maternal transmission during vaginal delivery is essential in acquiring these bacterial species". This statement should mention 'in early life' since it really is the early but not all the acquisition which is absolutely comparable between groups after month 1 (figure 3B).

We agree with the Reviewer that adding 'in early life' to our statement will help to clarify our meaning that maternal transmission during vaginal delivery is essential in acquiring *Bifidobacterium* spp. at this particular point in time. We have adjusted this accordingly at page 17, line 303 and page 20 line 371.

I feel that much of this part of the discussion is highly speculative as no clear health effects can be demonstrated in this study but a non-significant trend given the limited number of included subjects.

We agree with the Reviewer that the evidence is modest. However, we still found it important to mention the potential link between mode of delivery and health effects. We hope that by further adjusting our discussion as per the Reviewer's suggestion, we have appropriately underlined, but not overstated, the possible causal link between delivery mode-induced gut microbiota changes and health characteristics. Please see page 19 lines 343-351 and 362-364 for the nuances applied to our discussion.

Reviewer #4

I concur with the sentiment of Reviewer 1 that this paper discusses a very interesting and important issue. However, I also agree with Reviewer 2 that it is a little alarming that no analysis protocol (of any form) was pre-specified by the authors, and that an extremely large number of analyses have been performed. Nonetheless, I sympathise with the authors that this is not regular practice in their field, and that it may also be difficult to achieve. Furthermore, I found the newly provided flow diagrams on the performed analyses to be very clear, and I feel that the authors have addressed the previous reviewers' criticisms effectively. My comments therefore are mostly minor, seeking to address a small number of additional issues that I identified.

We thank the Reviewer for his/her comments and are pleased the flow diagrams have clarified our data analysis approach.

Specific comments

No justification appears to be provided by the authors as to why the number of RI events was dichotomised in to 0-2 and 3-7. This is troubling as it is easy to envisage that the presented results (which are discussed at length in the manuscript) may not be present if 0-3 and 4-7 was chosen instead. How was this categorisation chosen?

We thank the Reviewer for this comment, since it is indeed important to clarify that the dichotomization was based on objective criteria. The categorization of RI events was based on the results of previously published manuscripts from our research group, reporting on the respiratory microbiota development of this same birth cohort.^{1,2} The total number of RIs in this cohort ranges from 0-7, and based on the normal distribution the population was initially stratified into three groups when studying the respiratory microbiota maturation, namely 0-2, 3-4 and 5-7 RIs.¹ In a previously published paper, we observed that the 3-4 and especially the 5-7 RIs groups showed an altered respiratory microbiota maturation process, with decreased microbial community stability when compared to the 0-2 RIs group. Following, the group of children with 0-2 RIs was defined as the healthy reference group. In a more in depth study, analyzing the respiratory and oral niches of this birth cohort, a further dichotomization of RIs was used, again using the children with up 0-2 RIs as healthy reference group, versus children with >2 RIs over the first year of life as a more susceptible group.² The healthy reference group had a significantly better topographical differentiation between nasopharynx and oropharynx from 8 weeks on, compared to children who eventually suffered from >2 RIs over the first year of life. Based on these two previous studies of the same birth cohort, we decided to continue using the same stratification for reasons of consistency, and, as mentioned before, according to the normal distribution of RIs in the cohort. We have now underlined that

the categorization of our cohort is based on previous studies at page 14, lines 234-238 and page 27 lines 523-525 of our revised manuscript.

Clarification should be added to the online supplementary methods on how p-values were calculated for analyses performed using linear mixed models. I presume if nlme was used, that these are the default returned p-values that assume a denominator DoF equal to a corresponding balanced ANOVA design? (c.f., lme4 no longer returns p-values – this is not in general a simple issue.)

Indeed, we used the lme function of the nlme package and to calculate p-values of the linear mixed models we used the anova function. We have now added this to the supplementary methods at page 25, lines 495-496.

There are several ways in which I feel the presented figures could be improved:

Where at all possible, colour-blind friendly palettes should be used.

We agree with the Reviewer that the choice of a green-red color scheme was not ideal and have therefore now implemented colorblind-friendly palettes in all figures where datapoints overlap, to improve the visualization of group differences. Additionally, we used package *dichromat*³ to simulate the effects of different types of color-blindness (protanopia, deuteranopia and tritanopia) and found that the segments of our relative abundance bars are distinguishable in these cases. Finally, we ensured that all our figures are also gray-scale printer-friendly.

In the legends of figures that contain ellipses, clarification on the reason for the location of the centre point should be added (not just detail that the ellipse shape is dependent on standard deviations).

The location of the center point of the ellipses in the nMDS plots was calculated using the mean of the coordinates per group. We have now specified this in the relevant figure legends.

Superscripts and subscripts are now easy to incorporate, so text such as "R2" should be corrected.

We have changed R2 to R^2 in the relevant figures.

In Figure 4, there are labels for which no point can be seen, and labels that are obscured behind points; this should be resolvable by tweaking parameters in ggrepel.

We thank the Reviewer for this useful comment. We have now ensured that all labels are visible. We have added asterisks to the two OTUs of which the datapoints fall beyond the limits of the plot, namely *Eggerthella* and *Erysipelotrichaceae*, and have noted the corresponding log₂FC and adjusted p-values in the legend.

In Supplementary Figure 2, consider using jittering and reducing the length of the horizontal portion of the error bars in order to limit overlay.

We thank the Reviewer for his insightful feedback. We have adjusted the figure accordingly and agree this helps to better visualize the data.

Lines 114-115: I would clarify whether the cases with pre-/intrapartum antibiotic use were included in the analyses.

Yes, the two cases where pre-/intrapartum maternal antibiotics were given were included in the analyses. We have clarified this in lines 125-127 on page 10 of our revised manuscript.

Lines 131-133: I would add the formal statistical justification to back up the statement that the infant microbial community composition developed towards an adult-like profile.

We thank the Reviewer for this comment. The statement that the infants' overall microbial community composition developed slowly towards an adult-like profile was based on the visualization in Figure 1A. Here it can be seen that the composition of the children's samples approaches that of the mothers' over the course of the first year. As suggested by the Reviewer, we have now performed groupwise comparisons between children's and mothers' samples per timepoint as formal statistical justification to back up this statement. We calculated the Bray Curtis (BC) dissimilarity index between all children's and mothers' samples per timepoint, and found that the median BC dissimilarity decreased as time progressed (Table R1). We have summarized these results in the manuscript at page 11, lines 144-146 and have added the table as Supplementary Table 1.

Table R1. Decrease in BC dissimilarity between children's and mothers' samples over time

Timepoint	median BC dissimilarity	IQR25	IQR75
Post-partum	0.999	0.996	1.000
Day 1	0.986	0.903	0.998
Week 1	0.982	0.922	0.996
Week 2	0.971	0.895	0.993

Month 1	0.954	0.864	0.985
Month 2	0.935	0.845	0.978
Month 4	0.923	0.837	0.971
Month 6	0.904	0.807	0.966
Month 9	0.834	0.725	0.924
Month 12	0.739	0.633	0.839

Overview of groupwise BC dissimilarity indices between children's and mothers' samples per timepoint. IQR = interquartile range.

Line 202: "we would" not "we'd".

We have modified the text at page 13, lines 217-220.

HR on line 213: should this be 0.501 not -0.501?

We thank the Reviewer for this keen observation. Here, we mistakenly noted the beta coefficient of the model, where -0.501 indicates that VD children had a lower risk of getting prescribed antibiotics in the first year of life compared to CS children. The hazard ratio of our model is actually 0.606, indicating that vaginal birth reduces the chance of antibiotic prescription by 40% (p=0.134). We have corrected this in our revised manuscript at page 14, line 229.

Line 275: RIs not respiratory infections.

We have changed the text accordingly at page 17, line 296.

References

1. Bosch, A. A. T. M. *et al.* Maturation of the Infant Respiratory Microbiota, Environmental Drivers, and Health Consequences. A Prospective Cohort Study. *Am. J. Respir. Crit. Care Med.* **196**, 1582–1590 (2017).
2. Man, W. H. *et al.* Loss of Microbial Topography between Oral and Nasopharyngeal Microbiota and Development of Respiratory Infections Early in Life. *Am. J. Respir. Crit. Care Med.* rccm.201810-1993OC (2019). doi:10.1164/rccm.201810-1993OC
3. Thomas Lumley. dichromat: Color Schemes for Dichromats. (2013). Available at: <https://cran.r-project.org/package=dichromat>. (Accessed: 23rd July 2019)

REVIEWERS' COMMENTS:

Reviewer #2 (Remarks to the Author):

no further comments

Reviewer #4 (Remarks to the Author):

Title: Impact of delivery mode-associated gut microbiota dynamics on health in the first year of life

Author(s): Reyman M, van Houten MA, van Baarle D et al

Manuscript ID: NCOMMS-19-11133A

Date review requested: August 20 2019

Date Review Submitted: August 23 2019

Reviewer: Michael Grayling, Newcastle University

0. Summary

In my previous review of this manuscript, I expressed a small number of concerns regarding its content. In particular, these related to the method of dichotomisation of a subset of the data and the composition of a number of the figures.

In this revised manuscript, I commend the authors for having addressed each of these concerns in a concise but comprehensive manner. I have no additional comments to make and therefore now recommend the manuscript for publication.

Reviewer #2

No further comments.

We are glad the previous issues have now been resolved in our revised manuscript.

Reviewer #4

In my previous review of this manuscript, I expressed a small number of concerns regarding its content. In particular, these related to the method of dichotomisation of a subset of the data and the composition of a number of the figures.

In this revised manuscript, I commend the authors for having addressed each of these concerns in a concise but comprehensive manner. I have no additional comments to make and therefore now recommend the manuscript for publication.

We thank the Reviewer for the positive comments.